# Optimal rates for k-NN density and mode estimation

**Sanjoy Dasgupta**
University of California, San Diego, CSE
dasgupta@eng.ucsd.edu

**Samory Kpotufe** *
Princeton University, ORFE
samory@princeton.edu

## Abstract

We present two related contributions of independent interest: (1) high-probability finite sample rates for $k$-NN density estimation, and (2) practical mode estimators – based on $k$-NN – which attain minimax-optimal rates under surprisingly general distributional conditions.

## 1   Introduction

We prove finite sample bounds for $k$-nearest neighbor ($k$-NN) density estimation, and subsequently apply these bounds to the related problem of mode estimation. These two main results, while related, are interesting on their own.

First, $k$-NN density estimation [1] is one of the better known and simplest density estimation procedures. The estimate $f_k(x)$ of an unknown density $f$ (see Definition 1 of Section 3) is a simple functional of the distance $r_k(x)$ from $x$ to its $k$-th nearest neighbor in a sample $X_{[n]} \triangleq \{X_i\}_{i=1}^n$. As such it is intimately related to other functionals of $r_k(x)$, e.g. the degree of vertices $x$ in $k$-NN graphs and their variants used in modeling communities and in clustering applications (see e.g. [2]).

While this procedure has been known for a long time, its convergence properties are still not fully understood. The bulk of research in the area has concentrated on establishing its asymptotic convergence, while its finite sample properties have received little attention in comparison. Our finite sample bounds are concisely derived once the proper tools are identified. The bounds hold with high probability, under general conditions on the unknown density $f$. This generality proves quite useful as shown in our subsequent application to the problem of mode estimation.

The basic problem of estimating the modes (local maxima) of an unknown density $f$ has also been studied for a while (see e.g. [3] for an early take on the problem). It arises in various unsupervised problems where modes are used as a measure of typicality of a sample $X$. In particular, in modern applications, mode estimation is often used in clustering, with the modes representing cluster centers (see e.g. [4, 5] and general applications of the popular *mean-shift* procedure).

While there exists a rich literature on mode estimation, the bulk of theoretical work concerns estimators of a single mode (highest maximum of $f$), and often concentrates on procedures that are hard to implement in practice. Given the generality of our first result on $k$-NN density estimation, we can prove that some simple implementable procedures yield optimal estimates of the modes of an unknown density $f$, under surprisingly general conditions on $f$.

Our results are overviewed in the following section, along with an overview of the rich literature on $k$-NN density estimation and mode estimation. This is followed by our theoretical setup in Section 3; our rates for $k$-NN density estimation are detailed in Section 4, while the results on mode estimation are given in Section 5.

## 2 Overview of results and related Work

### 2.1 Rates for $k$-NN density estimates

The $k$-NN density estimator dates back perhaps to the early work of [1] where it is shown to be consistent when the unknown density $f$ is continuous on $\mathbb{R}^d$. While one of the best known and simplest procedure for density estimation, it has proved more cumbersome to analyze than its smooth counterpart, the kernel density estimator.

More general consistency results such as [6, 7] have been established since its introduction. In particular [6] shows that, for $f$ Lipschitz in a neighborhood of a point $x$, where $f(x) > 0$, and $k = k(n)$ satisfying $k \to \infty$ and $k/n^{2/(2+d)} \to 0$, the estimator is asymptotically normal, i.e. $\sqrt{k}(f_k(x) - f(x))/f(x) \xrightarrow{\mathcal{D}} \mathcal{N}(0, 1)$. The recent work of [8], concerning generalized *weighted* variants of $k$-NN, shows that asymptotic normality holds under the weaker restriction $k/n^{4/(4+d)} \to 0$ if $f$ is twice differentiable at $x$.

Asymptotic normality as stated above yields some insight into the rate of convergence of $f_k$: we can expect that $|f_k(x) - f(x)| \lesssim f(x)/\sqrt{k}$ under the stated conditions on $k$. In fact, [8] shows that such a result can be obtained in expectation for $n = n(x)$ sufficiently large. In particular, their conditions on $k$ allows for a setting of $k \approx n^{4/(4+d)}$ (not allowed under the above conditions) yielding a minimax-optimal $l_2$ risk $\mathbb{E} |f_k(x) - f(x)|^2 \lesssim f(x)^2/k = O(n^{-4/(4+d)})$.

While consistency results and bounds on expected error are now well understood, we still don't have a clear understanding of the conditions under which high probability bounds on $|f_k(x) - f(x)|$ are possible. This is particularly important given the inherent instability of nearest neighbors estimates which are based on order-statistics rather than the more stable average statistics at the core of kernel-density estimates. The recent result of [9] provides an initial answer: they obtain a high-probability bound uniformly over $x$ taking value in the sample $X_{[n]}$, however under conditions not allowing for optimal settings of $k$ (where $f$ is assumed Lipschitz).

The bounds in the present paper hold with high-probability, simultaneously for all $x$ in the support of $f$. Rather than requiring smoothness conditions on $f$, we simply give the bounds in terms of the modulus of continuity of $f$ at any $x$, i.e. how much $f$ can change in a neighborhood of $x$. This allows for a useful degree of flexibility in applying these bounds. In particular, optimal bounds under various degrees of smoothness of $f$ at $x$ easily follow. More importantly, for our application to mode estimation, the bounds allow us to handle $|f_k(x) - f(x)|$ at different $x \in \mathbb{R}^d$ with varying smoothness in $f$. As a result we can derive minimax-optimal mode estimation rates for practical procedures under surprisingly weak assumptions.

### 2.2 Mode estimation

There is an extensive literature on mode estimation and we unfortunately can only overview some of the relevant work. Most of the literature covers the case of a unimodal distribution, or one where there is a single maximizer $x_0$ of $f$.

Early work on estimating the (single) mode of a distribution focused primarily on understanding the consistency and rates achievable by various approaches, with much less emphasis on the ease of implementation of these approaches. The common approaches consist of estimating $x_0$ as $\hat{x} \triangleq \arg \sup_{x \in \mathbb{R}^d} f_n(x)$ where $f_n$ is an estimate of $f$, usually a kernel density estimate. Various work such as [3, 10, 11] establish consistency properties of the approach and achievable rates under various Euclidean settings and regularity assumptions on the distribution $\mathcal{F}$. More recent work such as [12, 13] address the problem of optimal choice of bandwidth and kernel to adaptively achieve the minimax risk for mode estimation. Essentially, under smoothness $\kappa$ (e.g. $f$ is $\kappa$ times differentiable), the minimax risk ($\inf_{\hat{x}} \sup_f \mathbb{E}_f \|\hat{x} - x_0\|$) is of the form $n^{-(\kappa-1)/(2\kappa+d)}$, as independently established in [14] and [15].

As noticed early in [16], the estimator $\arg \sup_{x \in \mathbb{R}^d} f_n(x)$, while yielding much insight into the problem, is hard to implement in practice. Hence, other work, apparently starting with [16, 14] have looked into so-called *recursive estimators of the (single) mode* which are practical and easy to update as the sample size increases. These approaches can be viewed as some form of gradient-

ascent of $f_n$ with carefully chosen step sizes. The later versions of [14] are shown to be minimax-optimal. Another line of work is that of so-called *direct mode estimators* which estimate the mode from practical statistics of the data [17, 18]. In particular, [18] shows that the simple and practical estimator $\arg\max_{x \in X_{[n]}} f_n(x)$, where $f_n$ is a kernel-density estimator, is a consistent estimator of the mode. We show in the present paper that $\arg\max_{x \in X_{[n]}} f_k(x)$, where $f_k$ is a $k$-NN density estimator, is not only consistent, but converges at a minimax-optimal rate under surprisingly mild distributional conditions.

The more general problem of estimating all modes of distribution has received comparatively little attention. The best known practical approach for this problem is the *mean-shift* procedure and its variants [19, 4, 20, 21], quite related to recursive-mode-estimators, as they essentially consist of gradient ascent of $f_n$ starting from every sample point, where $f_n$ is required to be appropriately smooth to ascend (e.g. a smooth kernel estimate). While mean-shift is popular in practice, it has proved quite difficult to analyze. A recent result of [22] comes close to establishing the consistency of mean-shift, as it establishes the convergence of the procedure to the right *gradient lines* (essentially the ascent path to the mode) if it is seeded from fixed starting points rather than the random samples themselves. It remains unclear however whether mean-shift produces only *true* modes, given the inherent variability in estimating $f$ from sample. This question was recently addressed by [23] which proposes a hypothesis test to detect *false* modes based on confidence intervals around Hessians estimated at the modes returned by any procedure.

Interestingly, while a $k$-NN density estimate $f_k$ is far from smooth, in fact not even continuous, we show a simple practical procedure that identifies any mode of the unknown density $f$ under mild conditions: we mainly require that $f$ is well approximated by a quadratic in a neighborhood of each mode. Our finite sample rates (on $\|\hat{x} - x_0\|$, for an estimate $\hat{x}$ of any mode $x_0$) are of the form $O(k^{-1/4})$, hold with high-probability and are minimax-optimal for an appropriate choice of $k = \Theta(n^{4/(4+d)})$.

If in addition $f$ is Lipschitz or more generally Hölder-continuous (in principle uniform continuity of $f$ is enough), all the modes returned above a level set $\lambda$ of $f_k$ can be optimally assigned to separate modes of the unknown $f$. Since $\lambda \xrightarrow{n \to \infty} 0$, the procedure consistently prunes false modes. This feature is made intrinsic to the procedure by borrowing from insights of [9, 24] on identifying false clusters by inspecting levels sets of $f_n$. These last works concern the related area of level set estimation, and do not study mode estimation rates.

As alluded to so far, our results are given in terms of local assumptions on modes rather than global distributional conditions. We show that any mode that is sufficiently *salient* (this is locally parametrized) w.r.t. the finite sample size $n$, is optimally estimated, while false modes are pruned away. In particular our results allow for $f$ having a countably infinite number of modes.

## 3   Preliminaries

Throughout the analysis, we assume access to a sample $X_{[n]} = \{X_i\}_{i=1}^n$ drawn i.i.d. from an absolutely continuous distribution $\mathcal{F}$ over $\mathbb{R}^d$, with Lebesgue-density function $f$. We let $\mathcal{X}$ denote the support of the density function $f$.

The $k$-NN density estimate at a point $x$ is defined as follows.

**Definition 1** ($k$-NN density estimate). *For every $x \in \mathbb{R}^d$, let $r_k(x)$ denote the distance from $x$ to its $k$-th nearest neighbor in $X_{[n]}$. The density estimate is given as:*

$$f_k(x) \triangleq \frac{k}{n \cdot v_d \cdot r_k(x)^d},$$

*where $v_d$ denotes the volume of the unit sphere in $\mathbb{R}^d$.*

All *balls* considered in the analysis are closed Euclidean balls of $\mathbb{R}^d$.

# 4 $k$-NN density estimation rates

In this section we bound the error in estimating $f(x)$ as $f_k(x)$ at every $x \in \mathcal{X}$. The main results of the section are Lemmas 3 and 4. These lemmas are easily obtained given the right tools: uniform concentration bounds on the empirical mass of balls in $\mathbb{R}^d$, using *relative* Vapnik-Chervonenkis bounds, i.e. Bernstein's type bounds rather than Chernoff type bounds (see e.g. Theorem 5.1 of [25]). We next state a form of these bounds for completion.

**Lemma 1.** *Let $\mathcal{G}$ be a class of functions from $\mathcal{X}$ to $\{0, 1\}$ with VC dimension $d < \infty$, and $\mathbb{P}$ a probability distribution on $\mathcal{X}$. Let $\mathbb{E}$ denote expectation with respect to $\mathbb{P}$. Suppose $n$ points are drawn independently at random from $\mathbb{P}$; let $\mathbb{E}_n$ denote expectation with respect to this sample. Then for any $\delta > 0$, with probability at least $1 - \delta$, the following holds for all $g \in \mathcal{G}$:*

$$ - \min(\beta_n \sqrt{\mathbb{E}_n g}, \beta_n^2 + \beta_n \sqrt{\mathbb{E}g}) \ \leq \ \mathbb{E}g - \mathbb{E}_n g \ \leq \ \min(\beta_n^2 + \beta_n \sqrt{\mathbb{E}_n g}, \beta_n \sqrt{\mathbb{E}g}), $$

*where $\beta_n = \sqrt{(4/n)(d \ln 2n + \ln(8/\delta))}$.*

These sort of relative VC bounds allows for a tighter relation (than Chernoff type bounds) between empirical and true mass of sets ($\mathbb{E}_n g$ and $\mathbb{E}g$) in those situations where these quantities are small, i.e. of the order of $\beta_n^2 = \tilde{O}(1/n)$ above. This is particularly useful since the balls we have to deal with are those containing approximately $k$ points, and hence of (small) mass approximately $k/n$.

A direct result of the above lemma is the following lemma of [26]. This next lemma essentially reworks Lemma 1 above into a form we can use more directly. We re-use $C_{\delta,n}$ below throughout the analysis.

**Lemma 2** ([26]). *Pick $0 < \delta < 1$. Let $C_{\delta,n} \triangleq 16 \log(2/\delta)\sqrt{d \log n}$. Assume $k \geq d \log n$. With probability at least $1 - \delta$, for every ball $B \subset \mathbb{R}^d$ we have,*

$$ \mathcal{F}(B) \geq C_{\delta,n} \frac{\sqrt{d \log n}}{n} \implies \mathcal{F}_n(B) > 0, $$

$$ \mathcal{F}(B) \geq \frac{k}{n} + C_{\delta,n} \frac{\sqrt{k}}{n} \implies \mathcal{F}_n(B) \geq \frac{k}{n}, \text{ and} $$

$$ \mathcal{F}(B) \leq \frac{k}{n} - C_{\delta,n} \frac{\sqrt{k}}{n} \implies \mathcal{F}_n(B) < \frac{k}{n}. $$

The main idea in bounding $f_k(x)$ is to bound the random term $r_k(x)$ in terms of $f(x)$ using Lemma 2 above. We can deduce from the lemma that if a ball $B(x, r)$ centered has mass roughly $k/n$, then its empirical mass is likely to be of the order $k/n$; hence $r_k(x)$ is likely to be close to the radius $r$ of $B(x, r)$. Now if $f$ does not vary too much in $B(x, r)$, then we can express the mass of $B(x, r)$ in terms of $f(x)$, and thus get our desired bound on $r_k(x)$ and $f_k(x)$ in terms of $f(x)$.

Our results are given in terms of how $f$ varies in a neighborhood of $x$, captured as follows.

**Definition 2.** *For $x \in \mathbb{R}^d$ and $\epsilon > 0$, define $\hat{r}(\epsilon, x) \triangleq \sup \left\{ r : \sup_{\|x - x'\| \leq r} f(x') - f(x) \leq \epsilon \right\}$, and $\check{r}(\epsilon, x) \triangleq \sup \left\{ r : \sup_{\|x - x'\| \leq r} f(x) - f(x') \leq \epsilon \right\}$.*

The continuity parameters $\hat{r}(\epsilon, x)$ and $\check{r}(\epsilon, x)$ (related to the modulus of continuity of $f$ at $x$) are easily bounded under smoothness assumptions on $f$ at $x$. Our high-probability bounds on the estimates $f_k(x)$ in terms of $f(x)$ and the continuity parameters are given as follows.

**Lemma 3** (Upper-bound on $f_k$). *Suppose $k \geq 4C_{\delta,n}^2$. Then, with probability at least $1 - \delta$, for all $x \in \mathbb{R}^d$ and all $\epsilon > 0$,*

$$ f_k(x) < \left( 1 + 2\frac{C_{\delta,n}}{\sqrt{k}} \right) (f(x) + \epsilon), $$

*provided $k$ satisfies $v_d \cdot \hat{r}(\epsilon, x)^d \cdot (f(x) + \epsilon) \geq \frac{k}{n} - C_{\delta,n} \frac{\sqrt{k}}{n}$.*

**Lemma 4** (Lower-bound on $f_k$). *Then, with probability at least $1 - \delta$, for all $x \in \mathbb{R}^d$ and all $\epsilon > 0$,*

$$f_k(x) \geq \left(1 - \frac{C_{\delta,n}}{\sqrt{k}}\right)(f(x) - \epsilon),$$

*provided $k$ satisfies $v_d \cdot \check{r}(\epsilon, x)^d \cdot (f(x) - \epsilon) \geq \frac{k}{n} + C_{\delta,n} \frac{\sqrt{k}}{n}$.*

The proof of these results are concise applications of Lemma 2 above. They are given in the appendix (long version). The trick is in showing that, under the conditions on $k$, there exists an $r \approx (k/(n \cdot f(x)))^{1/d}$ which is at most $\hat{r}(\epsilon, x)$ or $\check{r}(\epsilon, x)$ as appropriate; hence, $f$ does not vary much on $B(x, r)$ so we must have

$$\mathcal{F}(B(x,r)) \approx \text{volume}(B(x,r)) \cdot f(x) = v_d \cdot r^d \cdot f(x) \approx \frac{k}{n}.$$

Using Lemma 2 we get $r_k(x) \approx r$; plug this value into $f_k(x)$ to obtain $f_k(x) \approx (1 + 1/\sqrt{k})f(x)$.

Lemmas 3 and 4 allow a great deal of flexibility as we will soon see with their application to mode estimation. In particular we can consider various smoothness conditions simultaneously at different $x$ for different biases $\epsilon$.

Suppose for instance that $f$ is locally Hölder at $x$, i.e. $\exists r, L, \beta > 0$ s.t. for all $x' \in B(x, r)$, $|f(x) - f(x')| \leq L \|x - x'\|^\beta$. Then for small $\epsilon$, both $\hat{r}(\epsilon, x)$ and $\check{r}(\epsilon, x)$ are at least $(\epsilon/L)^{1/\beta}$; pick $\epsilon = O(f(x)/\sqrt{k})$ for $n$ sufficiently large, then by both lemmas we have, w.h.p., $|f_k(x) - f(x)| \leq O(f(x)/\sqrt{k})$ provided $k = \Omega(\log^2 n)$ and satisfies $v_d(1/L\sqrt{k})^{d/\beta}f(x) \geq Ck/n$ for some constant $C$. This allows for a setting of $k = \Theta\left(n^{2\beta/(2\beta+d)}\right)$ for a minimax-optimal rate of $|f_k(x) - f(x)| = O\left(n^{-\beta/(2\beta+d)}\right)$.

The ability to consider various biases $\epsilon$ would prove particularly helpful in the next section on mode estimation where we have to consider different approximations in different parts of space with varying smoothness in $f$. In particular, at a mode $x$, we will essentially have $\beta = 2$ ($f$ is twice differentiable) while elsewhere on $\mathcal{X}$ we might not have much smoothness in $f$.

# 5 Mode estimation

We start with the following definition of modes.

**Definition 3.** *We denote the set of modes of $f$ by $\mathcal{M} \equiv \{x : \exists r > 0, \forall x' \in B(x, r), f(x') < f(x)\}$.*

We need the following assumption at modes.

**Assumption 1.** *$f$ is twice differentiable in a neighborhood of every $x \in \mathcal{M}$. We denote the gradient and Hessian of $f$ by $\nabla f$ and $\nabla^2 f$. Furthermore, $\nabla^2 f(x)$ is negative definite at all $x \in \mathcal{M}$.*

Assumption 1 excludes modes at the boundary of the support of $f$ (where $f$ cannot be continuously differentiable). We note that most work on the subject consider only interior modes as we are doing here. Modes on the boundary can however be handled under additional boundary smoothness assumptions to ensure that $f$ puts sufficient mass on any ball around such modes. This however only complicates the analysis, while the main insights remain the same as for interior modes.

An implication of Assumption 1 is that for all $x \in \mathcal{M}$, $\nabla f$ is continuous in a neighborhood of $x$, with $\nabla f(x) = 0$. Together with $\nabla^2 f(x) \prec 0$ (i.e. negative definite), $f$ is well-approximated by a quadratic in a neighborhood of a mode $x \in \mathcal{M}$. This is stated in the following lemma.

**Lemma 5.** *Let $f$ satisfy Assumption 1. Consider any $x \in \mathcal{M}$. Then there exists a neighborhood $B(x, r), r > 0$, and constants $\hat{C}_x, \check{C}_x > 0$ such that, for all $x' \in B(x, r)$, we have*

$$\check{C}_x \|x' - x\|^2 \leq f(x) - f(x') \leq \hat{C}_x \|x' - x\|^2. \tag{1}$$

We can therefore parametrize a mode $x \in \mathcal{M}$ locally as follows:

**Definition 4** (Critical radius $r_x$ around mode $x$). *For every mode $x \in \mathcal{M}$, there exists $r_x > 0$, such that $B(x, r_x)$ is contained in a set $A_x$, satisfying the following conditions:*
*(i) $A_x$ is a connected component of a level set $\mathcal{X}^\lambda \triangleq \{x' \in \mathcal{X} : f(x') > \lambda\}$ for some $\lambda > 0$.*
*(ii) $\exists \hat{C}_x, \check{C}_x > 0, \forall x' \in A_x, \check{C}_x \|x' - x\|^2 \leq f(x) - f(x') \leq \hat{C}_x \|x' - x\|^2$. (So $A_x \cap \mathcal{M} = \{x\}$.)*

> Return $\arg\max_{x \in X_{[n]}} f_k(x)$.

Figure 1: Estimate the mode of a unimodal density $f$ from $X_{[n]}$.

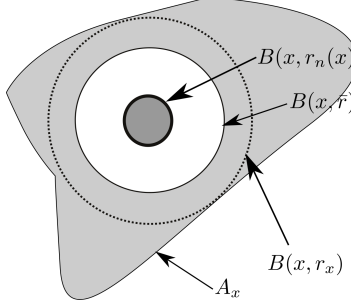

Figure 2: The analysis argues over different regions (depicted) around a mode $x$.

Finally, we assume that every hill in $f$ corresponds to a mode in $\mathcal{M}$:

**Assumption 2.** *Each connected component of any level set $\mathcal{X}^\lambda, \lambda > 0$, contains a mode in $\mathcal{M}$.*

### 5.1 Single mode

We start with the simple but common assumption that $|\mathcal{M}| = 1$. This case has been extensively studied to get a handle on the inherent difficulty of mode estimation. The usual procedures in the statistical literature are known to be minimax-optimal but are not practical: they invariably return the maximizer of some density estimator (usually a kernel estimate) over the entire space $\mathbb{R}^d$. Instead we analyze the practical procedure of Figure 1 where we pick the maximizer of $f_k$ out of the finite sample $X_{[n]}$. The rates of Theorem 1 are optimal ($O(n^{-1/(4+d)})$) for a setting of $k = O(n^{4/(4+d)})$.

**Theorem 1.** *Let $\delta > 0$. Assume $f$ has a single mode $x_0$ and satisfies Assumptions 1, 2. There exists $N_{x_0,\delta}$ such that the following holds for $n \geq N_{x_0,\delta}$. Let $\hat{C}_{x_0}, \check{C}_{x_0}$ be as in Definition 4. Suppose $k$ satisfies*

$$\left(\frac{24 C_{\delta,n} f(x_0)}{\check{C}_{x_0} r_{x_0}^2}\right)^2 \leq k \leq \left(\frac{1}{2}\sqrt{\frac{C_{\delta,n}}{\hat{C}_{x_0}}}\right)^{4d/(4+d)} f(x_0)^{(2d+4)/(4+d)} \left(\frac{v_d}{4} n\right)^{4/(4+d)}. \quad (2)$$

*Let $x$ be the mode returned in the procedure of Figure 1. With probability at least $1 - 2\delta$ we have*

$$\|x - x_0\| \leq 5\sqrt{\frac{C_{\delta,n}}{\check{C}_{x_0}} f(x_0)} \cdot \frac{1}{k^{1/4}}.$$

*Proof.* Let $r_{x_0}$ be the critical radius of Definition 4. Let $r_n(x_0) \equiv \inf \{r : B(x_0, r) \cap X_{[n]} \neq \emptyset\}$. Let $0 < \tau < 1$ to be later specified, and assume the event that $r_n(x_0) \leq \frac{\tau}{2} r_{x_0}$. We will bound the probability of this event once the proper setting of $\tau$ becomes clear.

Consider $\tilde{r}$ satisfying $r_{x_0} \geq \tilde{r} \geq 2r_n(x_0)/\tau$ (see Figure 2). We will first upper bound $f_k$ for any $x$ outside $B(x_0, \tilde{r})$, then lower-bound $f_k$ for $x \in B(x_0, r_n(x_0))$.

Recall $A_{x_0}$ from Definition 4. By equation (1) we have

$$\sup_{x \in A_{x_0} \setminus B(x_0, \tilde{r}/2)} f(x) \leq f(x_0) - \check{C}_{x_0} (\tilde{r}/2)^2 \triangleq \hat{F}. \quad (3)$$

The above allows us to apply Lemma 3 as follows. First note that for any $x \in \mathcal{X} \setminus B(x_0, \tilde{r}/2)$, $f(x) \leq \hat{F}$ since $A_{x_0}$ is a level set of the unimodal $f$, i.e. $\sup_{x \notin A_{x_0}} f(x) \leq \inf_{x \in A_{x_0}} f(x)$. Therefore, for any $x \in \mathcal{X} \setminus B(x_0, \tilde{r})$ let $\epsilon \doteq \hat{F} - f(x)$. By equation (3) the modulus of continuity $\hat{r}(\epsilon, x)$ is at least

Initialize: $\mathcal{M}_n \leftarrow \emptyset$.
For $\lambda = \max_{x \in X_{[n]}} f_n(x)$ down to 0:

- Let $\epsilon_\lambda \triangleq \lambda \cdot C_{\delta,n}/\sqrt{k}$.

- $\mathcal{M}_n \leftarrow \mathcal{M}_n \cup \left\{ x_i \triangleq \arg\max_{x \in \tilde{A}_i \cap X_{[n]}^\lambda} f_n(x) \right\}_{i=1}^m$.

Return the estimated modes $\mathcal{M}_n$.

Figure 3: Estimate the modes of a multimodal $f$ from $X_{[n]}$. The parameter $\tilde{\epsilon}$ serves to prune.

$\tilde{r}/2$. Therefore, if $k$ satisfies

$$v_d \cdot (\tilde{r}/2)^d \cdot \left( f(x_0) - \check{C}_{x_0}(\tilde{r}/2)^2 \right) \geq \frac{k}{n} - C_{\delta,n} \frac{\sqrt{k}}{n}, \tag{4}$$

we have with probability at least $1 - \delta$

$$\sup_{x \in \mathcal{X} \setminus B(x_0, \tilde{r})} f_k(x) < \left( 1 + 2\frac{C_{\delta,n}}{\sqrt{k}} \right) \left( f(x_0) - \check{C}_{x_0}(\tilde{r}/2)^2 \right). \tag{5}$$

Now we turn to $x \in B(x_0, r_n(x_0))$. We have again by equation (1) that $\inf_{x \in B(x, \tau\tilde{r})} f(x) \geq f(x_0) - \hat{C}_{x_0}(\tau\tilde{r})^2 \triangleq \check{F}$. Therefore, for $x \in B(x_0, r_n(x_0))$ let $\epsilon = f(x) - \check{F}$, we have $\check{r}(\epsilon, x) \geq \tau\tilde{r} - r_n(x_0) \geq \tau\tilde{r}/2$. It follows that, if $k$ satisfies

$$v_d \cdot ((\tau/2)\tilde{r})^d \cdot \left( f(x_0) - \hat{C}_{x_0}(\tau\tilde{r})^2 \right) \geq \frac{k}{n} + C_{\delta,n} \frac{\sqrt{k}}{n}, \tag{6}$$

we have by Lemma 4 that, with probability at least $1 - \delta$ (under the same event used in Lemma 3)

$$\inf_{x \in B(x, r_n(x_0))} f_k(x) \geq \left( 1 - \frac{C_{\delta,n}}{\sqrt{k}} \right) \left( f(x_0) - \hat{C}_{x_0}(\tau\tilde{r})^2 \right). \tag{7}$$

Next, with a bit of algebra, we can pick $\tau$ and $\tilde{r}$ so that the l.h.s. of (5) is less than the l.h.s. of equation (7). It suffices to pick $\tau^2 = \check{C}_{x_0}/8\hat{C}_{x_0}$ and $\tilde{r}^2 \geq 24 f(x_0) C_{\delta,n}/\check{C}_{x_0}\sqrt{k}$. Given these settings, equations (4) and (6) are satisfied whenever $k$ satisfies equation (2) of the lemma statement.

It follows that, with probability at least $1 - \delta$, $\inf_{x \in B(x, r_n(x_0))} f_k(x) > \sup_{x \in \mathcal{X} \setminus B(x_0, \tilde{r})} f_k(x)$. Therefore, the empirical mode chosen by the procedure is in $B(x_0, \hat{r})$. We are free to choose $\tilde{r}$ as small as $\max \left\{ \sqrt{24 f(x_0) C_{\delta,n}/\left( \check{C}_{x_0}\sqrt{k} \right)}, 2r_n(x_0)/\tau \right\}$.

We've assumed so far the event that $r_n(x_0) \leq \frac{\tau}{2} r_{x_0}$. We bound the probability of this event as follows. Let $r \triangleq \sqrt{24 f(x_0) C_{\delta,n}/\check{C}_{x_0}\sqrt{k}}$. Under the above setting of $\tau$, the Theorem's assumptions on $k$ imply that $r \leq r_{x_0}$, and that $v_d \cdot ((\tau/2)r)^d \cdot \left( f(x_0) - \hat{C}_{x_0}((\tau/2)r)^2 \right) \geq \frac{k}{n} + C_{\delta,n}\frac{\sqrt{k}}{n}$. Again, by equation (1), this implies that $\mathcal{F}(B(x_0, (\tau/2)r)) \geq \frac{k}{n} + C_{\delta,n}\frac{\sqrt{k}}{n}$. By Lemma, 2, with probability at least $1 - \delta$, $\mathcal{F}_n(B(x_0, (\tau/2)r)) \geq k/n$ and therefore $r_n(x_0) \leq (\tau/2)r \leq (\tau/2)r_{x_0}$. It now becomes clear that we can just pick $\tilde{r} = r$. □

## 5.2 Multiple modes

In this section we turn to the problem of estimating the modes of a more general density $f$ with an unknown number of modes.

The algorithm of Figure 3 operates on the following set of nested graphs $G(\lambda)$. These are subgraphs of a *mutual k-NN* graph on the sample $X_{[n]}$, where vertices are connected if they are in each other's nearest neighbor sets. The connected components (CCs) of these graphs $G(\lambda)$ are known to be good estimates of the CCs of corresponding level sets of the unknown density $f$ [9, 26, 27].

**Definition 5** (k-NN level set $G(\lambda)$)**.** *Given $\lambda \in \mathbb{R}$, let $G(\lambda)$ denote the graph with vertices in $X_{[n]}^{\lambda} \triangleq \left\{ x \in X_{[n]} : f_n(x) \geq \lambda \right\}$, and where vertices $x, x'$ are connected by an edge when and only when $\|x - x'\| \leq \alpha \cdot \min \{r_k(x), r_k(x')\}$, for some $\alpha \geq \sqrt{2}$.*

We will show that for a given $n$, any sufficiently salient mode is optimally recovered; furthermore, if $f$ is uniformly continuous on $\mathbb{R}^d$, then the procedure returns no false mode above a level $\lambda_n \to 0$.

### 5.2.1 Optimal Recovery for Any Mode

The guarantees of this section would be given in terms of *salient* modes as defined below. Essentially a mode $x_0$ is salient if it is separated from other modes by a sufficiently *wide* and *deep* valley. We define saliency in a way similar to [9], but simpler: we only require a *wide* valley since the smoothness of $f$ at the mode (as expressed in equation 1) takes care of the *depth*.

We start with a notion of separation between sets inspired from [26].

**Definition 6** (r-separation)**.** *$A, A' \subset \mathcal{X}$ are r-separated if there exists a (separating) set $S \subset \mathbb{R}^d$ such that: every path from $A$ to $A'$ crosses $S$, and $\sup_{x \in S + B(0,r)} f(x) < \inf_{x \in A \cup A'} f(x)$.*

Our notion of mode saliency follows: for a mode $x$, we require the critical set $A_x$ of Definition 4 to be well separated from all components at the level where it appears.

**Definition 7** (r-salient Modes)**.** *A mode $x$ of $f$ is said to be r-salient for $r > 0$ if the following holds. There exist $A_x$ as in Definition 4 (with the corresponding $r_x$, $\hat{C}_x$ and $\check{C}_x$), which is a CC of say $\mathcal{X}^{\lambda_x} \triangleq \{x \in \mathcal{X} : f(x) \geq \lambda_x\}$. $A_x$ is r-separated from $\mathcal{X}^{\lambda_x} \setminus A_x$.*

The next theorem again yields the optimal rates $O(n^{-1/(4+d)})$ for $k = O(n^{4/(4+d)})$.

**Theorem 2** (Recovery of salient modes)**.** *Assume $f$ satisfies Assumptions 1, 2. Suppose $\tilde{\epsilon} = \tilde{\epsilon}(n) \xrightarrow{n \to \infty} 0$. Let $x_0$ be an r-salient mode for some $r > 0$. Assume $k = \Omega\left(C_{\delta,n}^2\right)$. Then there exist $N = N\left(x_0, \{\tilde{\epsilon}(n)\}\right)$ depending on $x_0$ and $\tilde{\epsilon}(n)$ such that the following holds for $n \geq N$. Let $A_{x_0}, \hat{C}_{x_0}, \check{C}_{x_0}$ be as in Definition 4, and let $\lambda_{x_0} \triangleq \inf_{x \in A_{x_0}} f(x)$. Let $\delta > 0$. Suppose $k$ further satisfies*

$$\left( \frac{24 C_{\delta,n} f(x_0)}{\check{C}_{x_0} \min \left\{ r_{x_0}^2/4, (r/\alpha)^2 \right\}} \right)^2 \leq k \leq \left( \frac{1}{2} \sqrt{\frac{C_{\delta,n}}{\hat{C}_{x_0}}} \right)^{4d/(4+d)} \lambda_{x_0}^{(2d+4)/(4+d)} \left( \frac{v_d}{4} n \right)^{4/(4+d)}.$$

*Let $\mathcal{M}_n$ be the modes returned by the procedure of Figure 3. With probability at least $1 - 2\delta$, there exists $x \in \mathcal{M}_n$ such that*

$$\|x - x_0\| \leq 5 \sqrt{\frac{C_{\delta,n}}{\check{C}_{x_0}} f(x_0)} \cdot \frac{1}{k^{1/4}}.$$

### 5.2.2 Pruning guarantees

The proof of the main theorem of this section is based on Lemma 7.4 of [24].

**Theorem 3.** *Let $\Lambda \triangleq \sup_x f(x)$ and $r(\epsilon) \triangleq \sup_{x \in \mathbb{R}^d} \max \{\hat{r}(\epsilon, x), \check{r}(\epsilon, x)\}$. Assume $f$ satisfies Assumption 2. Suppose $r(\tilde{\epsilon}) = \Omega\left(k/n\right)^{1/d}$, which is feasible whenever $f$ is uniformly continuous on $\mathbb{R}^d$. In particular, if $f$ is Hölder continuous, i.e.*

$$\forall x, x' \in \mathbb{R}^d, \quad |f(x) - f(x')| \leq L \|x - x'\|^{\beta}, \text{ for some } L > 0, 0 < \beta \leq 1,$$

*then we can just let $\tilde{\epsilon} = \Omega\left(k/n\right)^{\beta/d}$ since $r(\tilde{\epsilon}) \geq (\tilde{\epsilon}/L)^{1/\beta}$. Define*

$$\lambda_0 = \max \left\{ 2\tilde{\epsilon}, 8\frac{\Lambda}{k} C_{\delta,n}^2, \left( \frac{k}{n} + C_{\delta,n} \frac{\sqrt{k}}{n} \right) \frac{2}{v_d r(\tilde{\epsilon})^d} \right\}.$$

*Assume $k \geq 9C_{\delta,n}^2$. The following holds with probability at least $1 - \delta$. Pick any $\lambda \geq 2\lambda_0$, and let $\lambda_f = \inf_{x \in X_{[n]}^{\lambda}} f(x)$. All estimated modes in $\mathcal{M}_n \cap X_{[n]}^{\lambda}$ can be assigned to distinct modes in $\mathcal{M} \cap \mathcal{X}^{\lambda_f}$.*

## Footnotes

*Much of this work was conducted when this author was at TTI-Chicago.

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
