[Reviews · NeurIPS 2014]

Submitted by Assigned_Reviewer_25

This paper presents the first finite sample analysis of k-NN density
estimation and mode estimation. The paper is well-written and the
results are novel and appear to be correct.

I think the results are quite impressive and, because mode estimation
plays an important role in density-based clustering, the results are
important.

My only negative comment is that there are no numerical experiments.
However, the theoretical results are substantial enough that this does
not affect the merit of the paper.

A few small comments for the authors:

-You give the impression that finding the modes of a kernel density estimator is
difficult. But it is easy if one uses the mean-shift algorithm.

-You should probably include this reference:
Romano, J. (1988). On weak convergence and optimality of kernel density estimates of the mode.
The Annals of Statistics, 16, 629-647.

-The first sentence of section 2.2 is missing a word.

-How do we choose alpha and tilde-epsilon in practice?

Summary: Great paper. Substantial new theoretical results.

Submitted by Assigned_Reviewer_29

The paper presents finite sample bounds for the error of k-NN density estimation and an optimal k-NN based computationally feasible mode estimation procedure. Both sets of results are stated largely in terms of local properties of the density near the points of interest. For density estimation, the key property is a modulus of continuity type quantity, which can easily be stated in terms of many of the commonly used (local or global) smoothness assumptions of the function. The results on mode estimation also depend on the quadractic behavior near the modes, as well as conditions to ensure minimal separation of the modes, both of which are in tune with other work on the topic.

I believe both sets of results will be of significant interest.

The optimal choice of k for both problems (and the parameter \tilde \epsilon for the multiple mode estimation procedure) depend on the unknown density. This is of course to be expected, but a few words on sane practical heuristics for their choice would be beneficial (even if it is to say that no such methods are known).

There appears to be something missing from the beginning of Lemma 4.
Summary: Due to the generality of the density estimation bounds and the computational simplicity of the mode estimation procedure, I believe the presented results are of significant interest.

Submitted by Assigned_Reviewer_35

This paper presents a finite-sample analysis of k-NN density estimation. The paper begins by establishing uniform high-probability bounds for sup_{x \in R^d} | fk(x) - f(x) |, where fk(x) is the density estimate and f(x) the true density. The paper then uses this result to show that k-NN density estimation induces mode estimates that converge at an optimal rate.

I enjoyed reading this paper, and found it to be clearly written. The technical development was well explained, making it possible to get a good feel for the main ideas used in the proof. The theoretical results also seem useful and relevant.

Minor comments:

- In Section 2.1, do any of the prior bounds hold simultaneously for all x in Rd? If not, it might be worth emphasizing this.
- Lemma 2, I'm not sure \mathcal{F} has been defined.
- Is there a reason lemmas 3 and 4 are not combined into a single result? To me, these two lemmas together form one of the most important results in the paper; I think if it were called a theorem it would be easier for the casual reader to take note.
- On line 319, you say that "A_{x_0} is a level set." It is not clear to me that this is true given the assumptions made? Unless I'm missing something, you did not assume f(x) to be continuous outside of Ax0, so f(x') could get arbitrarily close to f(x0) for some x' that is far from x0.
- The notation S + B(0, r) in Definition 6 should be defined.
- Finally, the expressions "e.g." and "i.e." need to be surrounded by commas on both sides, e.g., like just there.
Summary: Well written paper with relevant results.
Author Feedback
Author rebuttal: We thank all reviewers for their comments towards improving the paper. These
will all be taken into account. Please find answers to some questions below.

REVIEWER 25:
- We will further emphasize the existence of practical mode-estimation procedures such as mean-shift. These have however proved hard to analyze, and theoretical results have focused on less-practical (often non-implementable) procedures.
We aimed to show optimal bounds for simple implementable procedures.

- We will include the reference.

-We will comment further on the practical choices of parameters.
From the theory \alpha>sqrt{2} works, (\alpha =1 seems usually fine in light of other results) while setting \tilde{\epsilon} requires at least an upper-bound on f.

REVIEWER 29:
We will comment further on the practical choices of parameters.

REVIEWER 35:
-The prior bounds hold simultaneously for all x.
-We often call Lemmas 3 and 4 separately in the analysis of mode estimation. Otherwise they can be combined into a single theorem.
-A_{x_0} being a level set: this is extending Definition 4, and should (will) be made explicit in the statement of the Theorem.